# Associations between COVID-19-Related Digital Health Literacy and Online Information-Seeking Behavior among Portuguese University Students

**DOI:** 10.3390/ijerph17238987

**Published:** 2020-12-02

**Authors:** Rafaela Rosário, Maria R. O. Martins, Cláudia Augusto, Maria José Silva, Silvana Martins, Ana Duarte, Inês Fronteira, Neida Ramos, Orkan Okan, Kevin Dadaczynski

**Affiliations:** 1School of Nursing, University of Minho, 4710-057 Braga, Portugal; coliveira@ese.uminho.pt (C.A.); mjsilva@ese.uminho.pt (M.J.S.); aduarte@ese.uminho.pt (A.D.); 2Health Sciences Research Unit: Nursing (UICISA: E), Nursing School of Coimbra (ESEnfC), 3000-232 Coimbra, Portugal; silvana.martins12@gmail.com; 3Global Health and Tropical Medicine, Institute of Hygiene and Tropical Medicine, NOVA University of Lisbon, 1349-008 Lisboa, Portugal; mrfom@ihmt.unl.pt (M.R.O.M.); ifronteira@ihmt.unl.pt (I.F.); a21000772@ihmt.unl.pt (N.R.); 4Faculty of Educational Science, Interdisciplinary Centre for Health Literacy Research, Bielefeld University, 33615 Bielefeld, Germany; orkan.okan@uni-bielefeld.de; 5Department of Nursing and Health Science, Fulda University of Applied Sciences, 36037 Fulda, Germany; kevin.dadaczynski@pg.hs-fulda.de; 6Center for Applied Health Science, Leuphana University of Lüneburg, 21335 Lüneburg, Germany

**Keywords:** digital health literacy, COVID-19, university students, information-seeking behavior

## Abstract

We aim to evaluate the associations between digital health literacy (DHL) related to COVID-19 and online information-seeking behavior among university students. Methods: A total of 3.084 students (75.7% women), with an average age of 24.2 (SD = 7.5) participated in this cross-sectional study, most of whom (36.5%) were from social sciences and pursued a bachelor’s degree (50.7%). Data on COVID-19-related DHL and online information-seeking behavior were collected using an online questionnaire. Logistic regression models were performed. Results: As the pandemic progressed, participants showed a lower chance of achieving a sufficient DHL (OR = 0.7; 95% CI = 0.6; 0.9). Using search engines more often (e.g., Google) (OR = 0.7; 95% CI = 0.5; 0.9), Wikipedia (OR = 0.7; 95% CI = 0.6; 0.9) and social media (e.g., Facebook) (OR = 0.7; 95% CI = 0.6; 0.9) decreased the likelihood of achieving sufficient DHL related to COVID-19. More frequent use of websites of public bodies (OR = 1.7; 95% CI = 1.1; 2.5) increased the odds of reporting sufficient DHL. Conclusion: DHL is associated with university students’ online information-seeking behavior in the time of COVID-19. From a community and public health perspective, programs aiming at improving DHL should be highlighted.

## 1. Introduction

In recent years, health literacy (HL) has become more important and appealing for research and to public health practice and policy [1,2]. It has taken on greater importance within the European health agenda [3,4] as a resource for empowering citizens by enabling them to participate in health promotion activities. HL is considered a broad concept that can be applied to many health domains [5,6], and emphasizes people’s knowledge, motivation and competencies to access, understand, judge and integrate health information in everyday life [7]. Sufficient HL may mediate the effects of the social determinants of health disparities. Hence, strengthening HL, particularly in those with limited HL, may reduce disparities and promote greater equity in health [8]. A recent study on coronavirus and COVID-19-related HL in German adults (≥16 years) found 50.1% of the population to have difficulties in dealing with coronavirus and COVID-19-related health information in their everyday lives [9]. The greatest challenge was to judge whether media information on the coronavirus and COVID-19 was reliable, with almost half of the population reporting difficulties in this area. Nevertheless, HL is considered a multi-dimensional construct that develops over time [10], across different health contexts and through social interactions [11]. It is expected that people with improvements in HL over time may have more active involvement in health decisions and better health outcomes [8]. In this context, HL is an asset to be developed, an outcome of health education and communication [5].

In recent decades, both digitization and digital transformation have made great strides and have led to the high availability of digital and especially mobile devices and their everyday use. According to the Digital Economy and Society Index (DESI) 73% of the Portuguese use the internet at least once a week (compared to 85% in the EU as a whole). In contrast, the use of social networks is above the EU average (80% versus 65% in 2019) [12]. With the increasing availability and use of digital media, the importance of web-based health information is also growing substantially. In the context of online health information, the skills needed to search, select, judge, transform, communicate, and use health information [7] are summarized under the concepts of digital HL (DHL) [13]. It is known that a sufficient level of DHL is directly associated with the skills to use online information and health, and inversely associated with lack of those skills and adverse health outcomes [14,15]. Technological solutions and evidence-based and user-friendly online information may promote DHL, allowing access to information that is transparent, attractive, dynamic and personalized.

Currently, we are facing an unprecedented public and social health crisis. The emergence of the new coronavirus (SARS-CoV-2) and its transmission capacity has contributed to a situation of great uncertainty in the world population. Information about this pandemic outbreak is abundant and available through a full range of digital media and technologies, including mobile health apps, wearable technologies, online health services and tools [16]. For example, the biggest trend in Google search history is the new coronavirus, which has led to the development of a specific site with COVID-19 information [17]. In this context, people around the world were faced with complying with strict restrictions (e.g., stay at home order); however, as the outbreak progressed, the need to acquire more information not only about the disease but also about prevention and risk communication has become greater for people [18]. During the crisis a multitude of digital media (e.g., engines such as Google, Bing, Yahoo!, Wikipedia, social media, health portals) produce a plethora of information of varying quality, a phenomenon also known as an infodemic [19]. This emphasizes the critical need for HL and DHL. The uncertainty due to massive amounts of information is significantly higher among those with lower HL, females, those who have children, and those who are younger [9]. Research also shows that overabundance of information not only leads to confusion and uncertainty, but is also associated with health outcomes. In their study with Chinese adult citizens (≥18 years), Gao et al. [20] found that respondents with higher social media use were more often affected by adverse COVID-19 mental health outcomes such as fear.

Until now, there has been scarce research about the challenges that young adults, as so-called digital natives, experience when searching, finding, critically evaluating, transforming, communicating and using COVID-19-related information in everyday life. Moreover, there is a scant evidence about the determinants of DHL related to COVID-19. This investigation addresses these gaps, aiming to evaluate the associations between DHL related to COVID-19 and online information-seeking behavior among university students. Furthermore, we analyze the associations of DHL with the pandemic progression. Since the amount of COVID-19-related information and thus the probability of false and misinformation increases with the course of the pandemic, we hypothesized that students at the end of the survey, compared to those participating at the beginning, would have lower odds of achieving sufficient levels of DHL.

## 2. Materials and Methods

In this research, we aim to evaluate the associations between DHL related to COVID-19 and online information-seeking behavior in university students.

### 2.1. Participants

The present study is derived from the COVID-HL research consortium, a network composed of researchers from 49 countries conducting a survey on COVID-19-related DHL among university students (https://covid-hl.eu) [21]. A convenience sample of 3084 university students (75.7% of whom were female) participated in the Portuguese survey using the platform SurveyMonkey. Data collection occurred from 28 April (with 24,141 confirmed COVID-19 cases) to 8 June 2020 (with 34,885 confirmed COVID-19 cases). Most of the participating students were from the social sciences area (36.5%) and were pursuing a bachelor’s degree (50.7%). There were differences in mean age, type of course and studied degree (*p* < 0.001) between male and female students. Participating male students were older, predominantly from social and engineering sciences and pursued higher level degrees compared to female students (see Table 1).

### 2.2. HL Related to COVID-19

The study design and questionnaire were developed by Dadaczynski and colleagues [21] and are partly based on existing validated scales that have been adapted to the coronavirus/COVID-19 situation as well as newly developed scales for the research purposes of the COVID-HL network. The entire questionnaire is available on request from the last authors. The translation of the questionnaire to Portuguese followed several steps proposed by the WHO, including a forward translation, a panel of experts, backtranslation, pretesting, and creation of the final version [22]. Two researchers, with an in-depth understanding of the instrument and the population (university students), translated the instrument into Portuguese. The researchers’ group (panel of experts) comprised the translators and three researchers with experience in the field. The group reviewed the translation, looked for the integrity of the original instrument, in terms of semantic, conceptual, and technical equivalence, along with inconsistencies between the language of the original instrument and the translated document. Backtranslation was done by a bilingual person whose mother tongue is English and who had no information about the original English version. A group of university students was invited to pre-test the questionnaire, also providing in-depth comments on whether the questionnaire items were acceptable, understandable, and relevant in terms of style, content, and format. Our research group then examined the original and the back-translated documents and agreed on any significant differences when creating the final version.

The DHL instrument was based on a previously developed questionnaire with seven subscales of DHL [13]. We used five of the seven subscales, as two subscales seemed irrelevant to the target group of young adults (basic operational and navigation skills). Moreover, small adaptations were made to the original questions [13], to capture the specific (COVID-19) context of the study (e.g., “When you search the Internet for information on the coronavirus or related topics, how easy or difficult is it for you to…”). The questionnaire assessed the following dimensions of DHL: information searching, adding self-generated content, evaluating reliability, and determining relevance. Each DHL area was assessed with three items that could be answered on a 4-point Lickert scale ranging from 1 (very difficult) to 4 (very easy). A mean value was calculated for each subscale. For further analysis, two subgroups were created using the median split (sufficient versus limited DHL).

### 2.3. Online Information-Seeking Behavior

The sources used for online COVID-19 information-seeking were assessed by a self-developed item battery [23], including ten sources such as search engines (e.g., Google, Bing, Yahoo!), websites of public bodies, Wikipedia and social media (e.g., Facebook, Instagram, Twitter). Each of the online sources could be answered in relation to their frequency, including the following response categories: “do not know”, “never”, “rarely”, “sometimes”, and “often”. For all analyses, the response option “do not know” was excluded and the remaining response options were merged into two categories: “never and rarely” and “sometimes and often”.

### 2.4. Other Measurements

Subjective social status was based on the MacArthur Scale [24], which uses a ladder (1–10 points) as representing where people stand on the social hierarchy in Portugal. At the top of the ladder were the wealthiest people, with the highest education levels, and the most respected jobs (highest subjective social status). At the bottom were those with the least money, least education levels and the least respected jobs, or no job (lowest subjective social status).

Students were asked about their scientific field of study, according to the revised classification of science and technology (fos) of the Frascati manual [25]. Furthermore, they were asked about their pursued degree: bachelor’s, integrated master’s, master’s, PhD, and others.

### 2.5. Data Analysis

Descriptive statistics were used to explore item-specific normality, and participant characteristics are presented as means, standard deviations (SD), and percentages (%).

Confirmatory factor analysis (CFA) was used to assess the construct validity of the DHL-related instrument and to test a hypothesized model against previous theoretical and empirical reports in the literature [13]. We found that the current data adequately fit the DHL model, which was divided into the following four dimensions (12 items total): information searching (reasonable Cronbach’s alpha: α = 0.72); adding self-generated content (good Cronbach’s alpha: α = 0.85); evaluating reliability (weak Cronbach’s alpha: α = 0.65) and determining relevance (reasonable Cronbach’s alpha: α = 0.73). With the use of AMOS for the CFA, a review of the fit indexes revealed a chi-square/df value of 4.32, a goodness of fit index (GFI) value of 0.98, a comparative fit index (CFI) value of 0.98, and a root mean square error of approximation (RMSEA) value of 0.04. Furthermore, the chi-square test was significant χ^2^ (42) = 202.9; *p* < 0.001).

Bivariate differences were analyzed using Mann–Whitney U and chi-squared tests. Subsequently, associations between DHL related to COVID-19 (outcome variable) and sources used for COVID-19 information-seeking (predictor variable) were analyzed using logistic regression. An adjusted odds ratio (OR) with a 95% confidence interval (CI) was considered to determine the strength of association between DHL and its predictors of information sources used. This form of regression analysis was chosen because there are no predefined cut-off values for the outcome variable, resulting in an empirical division of sufficient versus limited DHL. As potential confounders, we included variables hypothesized as affecting DHL. We performed multivariate logistic regression, controlling for differences in students’ characteristics, including sex, age, subjective social status, course and degree of study. Data analyses were performed using SPSS, version 26.0 (IBM, SPSS Inc. Chicago, IL, USA), considering *p*-values of <0.05 as statistically significant.

### 2.6. Ethical and Legal Requirements

The study was designed and conducted in accordance with the Helsinki declaration and was approved by the Ethics Commission for Research in Life Sciences and Health (number CEICVS 020/2020). All students completed an informed written consent form before starting the survey.

## 3. Results

Concerning the four dimensions of DHL, male students reported significantly higher levels of DHL when compared with female in the dimensions adding-self generated content and evaluating reliability. Furthermore, male students reported significantly higher levels of DHL in two dimensions, adding self-generated content (χ^2^ (1) = 7.2, *p* = 0.007) and evaluating reliability (χ^2^ (1) = 4.8, *p* = 0.027) (see Table 2).

Regarding coronavirus and COVID-19 information-seeking behavior, and after adjusting for sex, age, subjective social status, course, and degree of study, students who searched “sometimes and often” in “search engines” had lower odds of achieving sufficient DHL with regard to “information searching” (OR = 0.7, 95% CI = 0.5; 0.9). Compared with students who reported to use “websites of public bodies” “never or rarely”, those who reported using specific online source “sometimes and often” had 1.7-fold (95% CI = 1.1; 2.5) and 1.6-fold (95% CI = 1.1; 2.5) increased odds of reporting sufficient DHL for the dimensions “evaluating reliability” and “determining the relevance”, respectively.

Students searching on “Wikipedia and other online-encyclopedias” more frequently had a lower odds of a sufficient DHL with regard to “evaluating the reliability” (OR = 0.7, 95% CI = 0.6; 0.9) and “determining the personal relevance” of the information obtained (OR = 0.7, 95% CI = 0.6; 0.9). Furthermore, using “social media” more frequently was associated with decreased odds of achieving sufficient DHL level for online “information searching” (OR = 0.7, 95% CI = 0.6; 0.9) and for the DHL dimension “evaluate the information reliability” (OR = 0.7, 95% CI = 0.6; 0.8). Finally, searching more frequently on health portals was associated with a sufficient level of “evaluating the reliability” of online health information (OR = 1.3, 95% CI = 1.1; 1.6) (see Table 3).

Compared with the first week of data collection, those students who participated in the survey during the last two weeks of data collection had a lower odds of achieving a sufficient action area “determining relevance” (OR = 0.7, 95% CI = 0.6; 0.9).

## 4. Discussion

To the best of our knowledge, this is the first cross-sectional study to investigate DHL, information-seeking behavior, and particularly preferred online information sources, in the time of coronavirus/COVID-19 among university students in Portugal.

For each dimension of DHL, university students tended to score mostly in the third and fourth quartile of the response range, meaning that they perceived their DHL as good to very good in relation to the given tasks and actions. With regard to the stratified analyses, the current study found significant gender differences in relation to DHL levels for the dimensions “adding self-generated content” and “evaluating reliability”. Compared to females, male students perceived fewer difficulties when adding their own digital content and evaluating the reliability of the health information they obtained from online sources. It is possible that female students were more self-critical regarding the information and hence scored lower on the evaluating subscale. However, no consistent pattern between sex and HL has been reported in the literature [26,27,28,29,30]. Furthermore, no conclusions can be drawn from this because gender is also related to the studied degree, and a former study found clear associations between respondents’ level of education and HL [31]. Furthermore, it is known that low HL is directly associated with a lower socioeconomic position [26,30,31]. Thus, we used a proxy measure of subjective social status [24], besides the studied degree and course, as potential confounders of the analysis.

Regarding the associations between DHL and online information-seeking behavior for coronavirus and COVID-19 information, we found similar trends among unadjusted and adjusted models (please see Appendix A). After adjusting for sex, age, subjective social status, course and degree of study, the DHL dimension “information searching” was inversely associated with using search engines and social media. In addition, students who used Wikipedia, other online encyclopedias and social media more frequently were less likely to achieve a sufficient level of DHL with regard to “assessing the reliability” of health information. A study on coronavirus and COVID-19-related HL in German adults found that respondents overall achieved a high level of HL (~50% of the participants) but reported to have difficulties in judging whether they could trust health information found online and on social media [9]. The interactive character of social media contributes to a breeding ground for misinformation, disinformation and conspiracy theories, which, in the worst case, can lead to harmful or risky behavior [32]. On the other hand, the results of a review also indicated that social media reduced worries and uncertainties when helpful information was provided [33]. Even if the evidence for COVID-19 is limited and very dynamic, it is recommended to manage the infodemic related to it, correcting false messages, e.g., by fact-checking, [34,35], delivering high quality information, using social inoculation or debunking. In addition, the providers of social media platforms are responsible for providing technical solutions for monitoring and screening misinformation, while journalists and editors should avoid inaccurate and misleading headlines.

Those who searched more frequently on public institution websites and health portals were more likely to achieve a sufficient level of DHL in assessing the reliability of health information. Moreover, the ability to determine the personal relevance of health information was positively associated with searching the websites of public bodies more often. The dimensions of appraising and applying health information (“evaluating reliability” and “determining relevance”) are considered more complex competencies, also named critical HL [5]. This indicates that searching on websites of public bodies and health portals, besides the likelihood of achieving sufficient HL skills, also leads to an advantage in relation to the more complex competences of appraising and applying information. Websites from public bodies are more expert-driven and hence provide reliable health information, which might increase subjective trustworthiness compared to other online information sources. This may result in positive effects as the recipients do not have to worry about the information’s reliability and can directly use this information in their everyday lives. This is in accordance with a two-sided concept, where HL is a relational and contextual concept that promotes and sustains effective health practices [7]. A recent study also emphasized that the release of authoritative information from professional agencies during the pandemic was an effective way to eliminate doubt [36].

Students who participated in the survey and completed the questionnaire in the last two weeks had a lower odd of achieving a sufficient level of HL in the dimension “determining relevance”. We believe this is an interesting result which points to the fact that, the longer the pandemic goes on for, the amount of information increases, which may lead to problems and challenges for people in finding and specifically applying appropriate information. Health information about the coronavirus and COVID-19 is available from various sources and perspectives [37,38], ranging from personal messages on social networks, to media and governments trying to support their communities and organizations. The amount of information has increased as the pandemic has progressed (also including dis/misinformation, and conspiracy theories), mainly because the knowledge on COVID-19 is considered incomplete and has shifted in a short period of time [39]. Appraising the most relevant and reliable information is considered profoundly difficult. Although people might become more experienced and familiarized with the information, they might be more self-critical and look for (new) information (self-doubt). This may result in stress, confusion and disorientation on the part of the recipients [36]. This calls for structural approaches to DHL and demands an interdisciplinary approach [39] in order to tackle the infodemic. This indicates a need to engage with the community as a whole, support people in selecting relevant information from the diverse and large amount of information available in order to inform their daily activities and encourage them to transfer and integrate this information into different situations and circumstances [7]. For example, media providers (e.g., social media) may offer tools and high-quality information content that allow information consumers to evaluate the source’s trustworthiness.

A limitation of this study that should be considered is its cross-sectional design, which does not establish causality among the variables. Furthermore, we used a random convenience sample of university students, which was not representative of Portuguese university students. University students are considered highly educated, which hampers the translation of these results to the general population. Finally, we used an online questionnaire which may exclude non internet users. Hence, the results may be only valid for those students who had access to the internet during the time of data collection.

The study also has important strengths. First, we acknowledge the novelty of the study. The pandemic and infodemic are highly prevalent and warrant studies like this, from different countries and disciplines. Addressing the singularities of the populations contributes to the advancement of intervention programs related to health promotion and prevention of COVID-19 [40], including health education measures to strengthen DHL and tackle the infodemic. Second, we emphasize that the current study is integrated into a network of DHL associated with COVID-19. Looking at the measurement properties of the DHL related to COVID-19, the overall reliability of each subscale of the instrument was considered sufficient, with satisfying Cronbach alpha scores and CFA in four action areas of DHL as added strengths. The results of a remaining subscale (protecting privacy) were less convincing, which indicates that it should be improved further. Finally, we performed analyses by adjusting for important confounders that were considered determinants of DHL.

## 5. Conclusions

This study supports media providers (e.g., social media) offering tools that allow information consumers to evaluate the source’s trustworthiness and provide technical possibilities for monitoring information availability. In addition, journalists and editors should avoid inaccurate and misleading headlines. It is recommended that the infodemic is managed using different strategies that respect the singularities of the target audience, as well as community engagement (whole of society approach). Hence, besides the information itself and the context, policymakers and end-users (in this case students) should also be integrated into intervention programs aimed at improving (digital) HL.

## Figures and Tables

**Table 1 ijerph-17-08987-t001:** Descriptives of participants.

	All	Females	Males	*p*
**Participants**	3084	75.7	23.9	
**Age [mean (SD)]**	24.2 (7.5)	23.8 (7.0)	25.5 (8.9)	≤0.001 ^a^
**Course [*n* (%)]**				
Engineering Sciences	386 (14.7)	195 (9.8)	188 (29.7)	≤0.001 ^b^
Humanities	145 (5.5)	109 (5.5)	34 (5.4)	
Exact sciences |natural| other	253 (9.6)	193 (9.7)	57 (9.0)	
Health sciences	886 (33.7)	752 (37.9)	134 (21.2)	
Social sciences, Psychology, Education	960 (36.5)	737 (37.1)	220 (34.8)	
**Degree of study [*n* (%)]**				
Bachelor	1331 (50.7)	1047 (52.8)	278 (43.9)	≤0.001 ^b^
Master (integrated)	544 (20.7)	373 (18.8)	168 (26.5)	
Post-graduation and master	543 (20.7)	407 (20.5)	135 (21.3)	
Doctorate	209 (8.0)	157 (7.9)	52 (8.2)	
**Subjective social status [*n* (%)]**				
Below median	1345 (51.4)	1020 (51.4)	325 (51.5)	0.967 ^b^
Median and above	1270 (48.6)	964 (48.6)	306 (48.5)	

^a^ Results from *t*-test or Mann–Whitney U; ^b^ Results from Qui squared test.

**Table 2 ijerph-17-08987-t002:** Digital health literacy (DHL) related to COVID-19 among female and male university students.

DHL Related to COVID-19 [Median (Interquartile Range)]	All	Females	Males	*p*
**Information Searching**				
Limited [*n* (%)]	999 (54.7)	766 (55.6)	233 (52.0)	0.2
Sufficient [*n* (%)]	827 (45.3)	612 (44.4)	215 (48.0)	
…make a choice from all the information you find?	3.0 (3.0; 4.0)	3.0 (3.0; 4.0)	3.0 (3.0; 4.0)	0.1
…use the proper words or search query to find the information you are looking for?	3.0 (3.0; 4.0)	3.0 (3.0; 4.0)	3.0 (3.0; 4.0)	0.2
… find the exact information you are looking for?	3.0 (3.0; 3.0)	3.0 (3.0; 3.0)	3.0 (3.0; 3.0)	0.5
**Adding self-generated content**				
Limited [*n* (%)]	1308 (72.1)	1009 (73.8)	299 (67.2)	0.007
Sufficient [*n* (%)]	505 (27.9)	359 (26.2)	146 (36.8)	
…clearly formulate your question or health-related worry?	3.0 (3.0; 3.0)	3.0 (3.0; 3.0)	3.0 (3.0; 3.0)	0.3
…express your opinion, thoughts, or feelings in writing?	3.0 (2.0; 3.0)	3.0 (2.0; 3.0)	3.0 (3.0; 4.0)	0.01
…write your message as such, for people to understand exactly what you mean?	3.0 (3.0; 3.0)	3.0 (3.0; 3.0)	3.0 (3.0; 4.0)	0.004
**Evaluating reliability**				
Limited [*n* (%)]	993 (54.4)	770 (55.9)	223 (49.9)	0.03
Sufficient [*n* (%)]	832 (45.6)	608 (44.1)	224 (50.1)	
...decide whether the information is reliable or not?	3.0 (3.0; 3.0)	3.0 (3.0; 3.0)	3.0 (3.0; 3.0)	0.1
…decide whether the information is written with commercial interests (eg. by people trying to sell a product)?	3.0 (3.0; 4.0)	3.0 (3.0; 3.0)	3.0 (3.0; 4.0)	0.2
…check different websites to see whether they provide the same information?	3.0 (3.0; 4.0)	3.0 (3.0; 4.0)	3.0 (3.0; 4.0)	0.2
**Determining relevance**				
Limited [*n* (%)]	970 (53.2)	725 (52.7)	245 (54.8)	0.4
Sufficient [*n* (%)]	854 (46.8)	652 (47.3)	202 (45.2)	
…decide if the information you found is applicable to you?	3.0 (3.0; 4.0)	3.0 (3.0; 4.0)	3.0 (3.0; 4.0)	0.3
…apply the information you found in your daily life?	3.0 (3.0; 4.0)	3.0 (3.0; 4.0)	3.0 (3.0; 4.0)	0.2
…use the information you found to make decisions about your health (eg. on protective measures, hygiene regulations, transmission routes, risks and their prevention)?	3.0 (3.0; 4.0)	3.0 (3.0; 4.0)	3.0 (3.0; 4.0)	0.04

Note: Sample sizes vary according to missing data, in students that did not answer the full questionnaire. *p*-value results from Qui squared test.

**Table 3 ijerph-17-08987-t003:** Associations between sources of information and DHL related to COVID-19.

	DHL Related to COVID-19 Adjusted OR (95% CI)			
	Information Searching	Adding Self-Generated Content	Evaluating Reliability	Determining Relevance
**Weeks of data collection**				
Week 1	ref	ref	ref	ref
Week 2	0.9 (0.7; 1.2)	1.0 (0.7; 1.3)	0.8 (0.7; 1.1)	0.9 (0.7; 1.2)
Week 3 and 4	0.7 (0.5; 1.1)	0.8 (0.5; 1.3)	0.7 (0.5; 1.1)	1.1 (0.7; 1.6)
Week 5 and 6	1.0 (0.8; 1.3)	0.8 (0.6; 1.1)	0.8 (0.6; 1.1)	**0.7 (0.6; 0.9)**
**Search engines (e.g., Google, Bing, Yahoo!)**				
Never and rarely	ref	Ref	Ref	ref
Sometimes and often	**0.7 (0.5; 0.9)**	0.8 (0.6; 1.1)	0.8 (0.6; 1.1)	0.8 (0.6; 1.1)
**Websites of public bodies**				
Never and rarely	ref	ref	ref	ref
Sometimes and often	1.4 (1.0; 2.2)	1.6 (1.0; 2.5)	**1.7 (1.1; 2.5)**	**1.6 (1.1; 2.5)**
**Wikipedia and other online-encyclopaedias**				
Never and rarely	Ref	ref	ref	Ref
Sometimes and often	0.8 (0.7; 1.0)	0.8 (0.7; 1.1)	**0.7 (0.6; 0.9)**	**0.7 (0.6; 0.9)**
**Social media (e.g., Facebook, Instagram, Twitter)**				
Never and rarely	Ref	ref	ref	Ref
Sometimes and often	**0.7 (0.6; 0.9)**	0.9 (0.7; 1.1)	**0.7 (0.6; 0.8)**	0.9 (0.7; 1.1)
**YouTube**				
Never and rarely	ref	Ref	Ref	Ref
Sometimes and often	1.0 (0.9; 1.2)	1.0 (0.8; 1,3)	0.9 (0.7; 1.1)	1.0 (0.9; 1.2)
**Blogs on health topics**				
Never and rarely	ref	Ref	Ref	Ref
Sometimes and often	0.9 (0.7; 1,2)	1.0 (0.8; 1,3)	0.9 (0.7; 1,1)	0.9 (0.7; 1,1)
**Guidebook-communities (e.g., SNS-24)**				
Never and rarely	ref	Ref	Ref	Ref
Sometimes and often	1.1 (0.8; 1.4)	1.1 (0.8; 1.4)	0.9 (0.7; 1.1)	0.9 (0.8; 1.2)
**Health portals**				
Never and rarely	ref	Ref	Ref	Ref
Sometimes and often	1.0 (0.8; 1.2)	1.1 (0.9; 1.4)	**1.3 (1.1; 1.6)**	1.1 (0.9; 1.3)
**Websites of doctors or health insurance companies**				
Never and rarely	ref	Ref	Ref	Ref
Sometimes and often	1.0 (0.8; 1.3)	1.1 (0.9; 1.4)	1.1 (0.9; 1.4)	1.1 (0.9; 1.4)
**News portals (e.g., of newspapers, TV stations)**				
Never and rarely	ref	ref	ref	ref
Sometimes and often	1.1 (0.8; 1.4)	1.0 (0.8; 1.4)	1.2 (0.9; 1.6)	1.1 (0.9; 1.5)

Note: Results from binary logistic regression *Odds ratio* (Confidence Interval). Model adjusted for sex, age, subjective social status, course and degree of study. Bold: *p* < 0.05.

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
