# Peer review of "Associations between COVID-19-Related Digital Health Literacy and Online Information-Seeking Behavior among Portuguese University Students"

_ijerph, 2020, doi:10.3390/ijerph17238987_

Round 1

Reviewer 1 Report

The article describes the barries to ability of individuals to synthesise and critically evaluate COVID-19 information available online. The methodology used is well described and could be replicated by other researchers. The article is interesting, and it is impressive the researchers were able to undertake this study so quickly during a time of considerable work and personal disruption.  The statements in the discussion seem well aligned with the results of the study, as does the concluding statement.

  • INTRODUCTION: The introduction is well written and provides the reader with a good background to the research. However, I would recommend including a definition of Digital Health Literacy. The emphasise of this research seems to be about how members of the general public synthesise and critically evaluate online health information. This is slightly different to digital health literacy, which is frequently used to describe an individual’s competency using digital technologies for preventative care, or the manage their health. As the use in this article is slightly different to that, a definition of what the authors are describing as Digital Health Literacy would help the reader.
  • METHODS: The methodology could be strengthened with a statement of aims or research questions before: 2.1 Participants.  
  • METHODS: It would be beneficial if the authors could include the final survey instrument as an appendix document.

Author Response

Response to Reviewer 1 Comments

Point 1: The article describes the barriers to ability of individuals to synthesise and critically evaluate COVID-19 information available online. The methodology used is well described and could be replicated by other researchers. The article is interesting, and it is impressive the researchers were able to undertake this study so quickly during a time of considerable work and personal disruption.  The statements in the discussion seem well aligned with the results of the study, as does the concluding statement.

 Response 1: We thank the reviewer.

Point 2: The introduction is well written and provides the reader with a good background to the research. However, I would recommend including a definition of Digital Health Literacy. The emphasise of this research seems to be about how members of the general public synthesise and critically evaluate online health information. This is slightly different to digital health literacy, which is frequently used to describe an individual’s competency using digital technologies for preventative care, or the manage their health. As the use in this article is slightly different to that, a definition of what the authors are describing as Digital Health Literacy would help the reader.

Response 2: We thank the reviewer for addressing this point. We clarified the definition of DHL in the introduction, please see lines 65-71 of the revised manuscript.

Point 3: The methodology could be strengthened with a statement of aims or research questions before: 2.1 Participants. 

Response 3: We thank the reviewer for the suggestion, which was included in the revised manuscript accordingly. Please see lines 101-102 of the revised version of the manuscript.

Point 4: It would be beneficial if the authors could include the final survey instrument as an appendix document.

Response 4: We thank the reviewer for the suggestion, the final questionnaire is available on request (in compliance with the policy of the COVID-HL consortium, see https://pub.uni-bielefeld.de/record/2942920). The following information was included in the methods section: “The entire questionnaire is available on request from the last authors”.

Reviewer 2 Report

The manuscript is timely.  With the onset of information regarding COVID-19 continually being disseminated in various formats, it is essential to have the skill to decipher credible digital health information.  Attached is the pdf with comments and suggested corrections.

Author Response

Response to Reviewer 2 Comments

Point 1: (In introduction) This is really hard to read. Consider splitting into two sentences.

Response 1: We thank the reviewer. We rephrased the sentence, please see lines 47-49 of the revised version of the manuscript.

We thank the reviewer for addressing all the English corrections. They were all integrated into the revised version of the manuscript.

Reviewer 3 Report

Dear authors, 

The manuscript offers important information from a large sample of students in Portugal about the association between digital health literacy related to COVID-19 and information seeking behaviors. 

There is one overarching limitation here that requires a major rethink of the study hypotheses and analysis – the proposed association between DHL and information behaviors. That is, the authors focus on factors related to ‘achieving a sufficient DHL’, which is in contrast to the conceptualization of health literacy as a stable orientation toward health information (that can be applied to a specific context, such as COVID). It makes more sense, building on research on information seeking as well as on health literacy, to test whether DHL predicts information seeking behaviors, and whether this varies by the source of information etc. This analysis would use DHL as a continuous independent variable (not binary) and would control for other possible predictors of information seeking, such as age, ethnicity, income, gender, education etc.

This requires an overhaul of much of the framework of the study and its hypotheses, as well as a revision of the analysis (depending on the outcome measure OLS may be possible, or perhaps logistic regression if the measure needs to be collapsed). As the study uses cross-sectional data, the issue of causal direction cannot be established, but it makes more sense that a stable individual measure of DHL would predict the likelihood and frequency of information seeking than the reverse. The results would then examine the magnitude and significance of the association between increases in DHL and information seeking outcomes.

Minor comment – the social status scale conflates education with income, when these should ideally be included as separate demographic control variables, prior to adding DHL to the model as the primary predictor.

Author Response

Response to Reviewer 3 Comments

Point 1: There is one overarching limitation here that requires a major rethink of the study hypotheses and analysis – the proposed association between DHL and information behaviors. That is, the authors focus on factors related to ‘achieving a sufficient DHL’, which is in contrast to the conceptualization of health literacy as a stable orientation toward health information (that can be applied to a specific context, such as COVID).

Response 1: We thank the reviewer for addressing the conceptualization of health literacy. Previous studies found thatlow or inadequate health literacy is considered a relatively stable patient characteristic, a risk that needs to be managed in the process of providing clinical care (1). Other definitions, as the reviewer described, present health literacy as a set of individual capacities, relatively stable over time, that allow the person to acquire and use new information (2). However, health literacy is also an action-oriented concept rather than simply an intellectual capacity. Health literacy is developed over time (3), across different health contexts and through social interactions (4). Hence, health literacy has leverage within the European health agenda (5, 6) as a resource for empowering citizens by enabling them to participate in health promotion activities. It is expected that people with improvements in health literacy over time, may have more active involvement in health decisions and better health outcomes (7). We clarified this conceptualization in the manuscript (lines 55-59 of introduction).

Point 2: It makes more sense, building on research on information seeking as well as on health literacy, to test whether DHL predicts information seeking behaviors, and whether this varies by the source of information etc.

Response 2: We thank the reviewer for the suggestion. We agree with the reviewer when (s)he says that this is a cross-sectional study, hence it is not our intention to establish causal relations. However, we believe that DHL must remain as an outcome variable. First, we are followed by the conceptualization addressed in the former point. Second, previous studies described two models of health literacy, the clinical risk and the personal asset (1). The first emphasizes the importance of communication and health service organization that is tailored to the needs of low literate individuals, and the latter model emphasizes health literacy as an asset to be developed, as an outcome of health education and communication (1). The asset model conceptualizes the promotion of health literacy beyond the transmission of health information, including more sophisticated insights into the potential of education and health communications (1), which comprises in the digital context the online sources of information. We clarified this position in the introduction (lines 55-59 and 92-93) and the statistical analysis (lines 179-180) of the revised version of the manuscript.

Point 3: This analysis would use DHL as a continuous independent variable (not binary) and would control for other possible predictors of information seeking, such as age, ethnicity, income, gender, education etc.
(…) This requires an overhaul of much of the framework of the study and its hypotheses, as well as a revision of the analysis (depending on the outcome measure OLS may be possible, or perhaps logistic regression if the measure needs to be collapsed).

Response 3: We thank the reviewer for this suggestion. Until now, we have no cut-off values available for the DHL variable, i.e. the division into sufficient versus limited was done using a median split. We used the median split because the questionnaire of DHL included only ordinal variables and the interpretation of the results with a continuous variable would be based on an artificial perspective. According to Andy Field (8) continuous variables may be continuous, i.e. a score that takes on any value on the measurement, and “discrete” when the score takes on only certain values (usually whole numbers) on a scale. Although the continuum exists underneath the scale of an ordinal variable (for example in a 4-point scale rating 2.53 makes sense), the actual and practical values that the variable takes on are limited. Therefore, we used binary logistic regression, which uses odds ratios to determine the likelihood of an outcome occurs. To allow a better understanding of our analysis, we included the following sentence in the revised of the manuscript: “This form of regression analysis was chosen because there are no predefined cut-off values for the outcome variable resulting in an empirical division of sufficient versus limited DHL.” (please see lines 182-184).

Point 4: Minor comment -The social status scale conflates education with income, when these should ideally be included as separate demographic control variables prior to adding DHL to the model as the primary predictor.

Response 4: We thank the reviewer for this suggestion. We included a supplementary file with the unadjusted odds ratio (Confidence Interval) for the social status scale and the information seeking behavior.

References

  1. Nutbeam D. The evolving concept of health literacy. Soc Sci Med. 2008;67(12):2072-8.
  2. Baker DW. The meaning and the measure of health literacy. J Gen Intern Med. 2006;21(8):878-83.
  3. Zarcadoolas C, Pleasant A, Greer DS. Elaborating a definition of health literacy: a commentary. J Health Commun. 2003;8 Suppl 1:119-20.
  4. Edwards M, Wood F, Davies M, Edwards A. The development of health literacy in patients with a long-term health condition: the health literacy pathway model. BMC Public Health. 2012;12:130.
  5. Visscher BB, Steunenberg B, Heijmans M, Hofstede JM, Devillé W, van der Heide I, et al. Evidence on the effectiveness of health literacy interventions in the EU: a systematic review. BMC Public Health. 2018;18(1):1414.
  6. Kickbusch I, Pelikan J, Apfel F, Tsouros A. Health literacy: the solid facts. Copenhagen, Denmark: WHO; 2013.
  7. Nutbeam D, Lloyd JE. Understanding and Responding to Health Literacy as a Social Determinant of Health. Annu Rev Public Health. 2020.
  8. Field A. Why is my evil lecturer forcing me to learn statistics.Discovering statistics using IBM SPSS statistics. 5th ed. Germany: SAGE; 2018. p. 1-45.

Round 2

Reviewer 3 Report

Thank you for clarifying your response to the suggestions.